# Evaluation of Effective Elastic Properties of Nitride NWs/Polymer Composite Materials Using Laser-Generated Surface Acoustic Waves

**Evgeny Glushkov** [1], **Natalia Glushkova** [1], **Bernard Bonello** [2], **Lu Lu** [3], **Eric Charron** [2], **Noëlle Gogneau** [3], **François Julien** [3], **Maria Tchernycheva** [3] and **Olga Boyko** [2,*]

[1] Institute for Mathematics, Mechanics and Informatics, Kuban State University, 350080 Krasnodar, Russia; evg@math.kubsu.ru (E.G.); nvg@math.kubsu.ru (N.G.)
[2] CNRS, Institut des NanoSciences de Paris (INSP), Sorbonne Universités, CEDEX 05, 75252 Paris, France; bernard.bonello@insp.jussieu.fr (B.B.); Eric.Charron@insp.jussieu.fr (E.C.)
[3] Centre de Nanosciences et de Nanotechnologies, UMR 9001 CNRS, Univ. Paris Sud, Univ. Paris-Saclay, 8 Avenue de la Vauve, 91120 Palaiseau, France; lu.lu@c2n.upsaclay.fr (L.L.); noelle.gogneau@c2n.upsaclay.fr (N.G.); francois.julien@c2n.upsaclay.fr (F.J.); maria.tchernycheva@u-psud.fr (M.T.)
[*] Correspondence: Olga.Boyko@insp.jussieu.fr; Tel.: +33-1-44-27-45-33

**Abstract:** In this paper we demonstrate a high potential of transient grating method to study the behavior of surface acoustic waves in nanowires-based composite structures. The investigation of dispersion curves is done by adjusting the calculated dispersion curves to the experimental results. The wave propagation is simulated using the explicit integral and asymptotic representations for laser-generated surface acoustic waves in layered anisotropic waveguides. The analysis of the behavior permits to determine all elastic constants and effective elastic moduli of constituent materials, which is important both for technological applications of these materials and for basic scientific studies of their physical properties.

**Keywords:** surface acoustic waves; GaN nanowires

## 1. Introduction

Semiconductor nanowires are nanostructures with a large aspect ratio and dimensions ranging typically between 5 nm and 100 nm in diameter and about 100 nm up to several micrometers in length. They are often referred to as one-dimensional nanostructures (1D). Due to a vast choice of materials from which they can be synthesized and the number of interesting physical properties that they possess, nanowires (NWs) have recently emerged as alternative building blocks for nanoscale electronics, optoelectronics as well as chemical and biological sensing at the molecular scale [1–4]. In particular, Gallium nitride (GaN) nanowires have attracted extensive research interest for their enhanced piezo-electric properties and nanoscale device applications [5].

Although many applications require knowledge and ability to control the mechanical behavior of NW based nanostructures, the elastic properties of such structures remain relatively unexplored. A few recent works focus mainly on individual NWs without considering a surrounding medium [6,7]. Measuring the mechanical properties of nanowires by conventional techniques is not trivial. Concerning individual NWs, optical measurements used commonly in microelectromechanical systems (MEMS) are not easily applicable to the NW resonators because the diameter is less than the visible wavelength. Atomic force microscopy (AFM) has been used to measure the Young modulus

of individual NWs [8,9]. Alternatively, transmission electron microscopy (TEM) permits direct and quantitative determination of mechanical resonances by applying an actuating signal between the nanowire and a counter-electrode [7]. Thus, the analysis of the resonant behavior of individual nanowires permits conclusions about their elastic constants. Studied by these two techniques, the values of the Young modulus published in the literature are controversial. Indeed, one part of the published studies shows that the Young modulus decreases by reducing the NW diameter [7,10], while others present the opposite behavior, i.e., the Young modulus increases with the NW diameter reduction [7,11]. Finally, some publications report the independence of the Young modulus from the wire diameter [12].

Meanwhile, the properties of individual nanowires may not provide reliable information about the mechanical properties of the nanowire-based devices. The integration of nanowires into devices often uses the polymer embedding the NW array. Thus, for device operation, it is important to consider not only individual stand-alone NWs, but also the surrounding medium (polymer), whose properties may vary (e.g., with annealing temperature and time). In this work, we focus on the elastic properties of nitride nanowires, which are promising materials for the fabrication of efficient and compact piezogenerators. Their tremendous piezoelectric and mechanical properties give them the ability to convert efficiently mechanical energy into electrical energy. Using an adapted AFM resiscope, Jamond et al. [5] showed the great potential of nitride nanowires for piezogeneration and the correlation between the polarity of the nanostructure, its deformation and the establishment of the piezopotential.

In this study, we use the prototype of a piezogenerator based on a vertical array of GaN nanowires. The wires are embedded in a polymer of Hydrogen silsesquioxane (HSQ), with their tops connected by a CrPt metal electrode providing Schottky contact. The effective elastic moduli of the fabricated HSQ-NW composite medium are determined via the minimization of the discrepancy between the calculated and measured characteristics of the surface acoustic waves (SAWs). This minimization is performed by varying of the input sample's material constants within the computer model used for SAW simulation. To excite and measure the SAWs, we use the Transient Grating Method (TGM) based on the exciting and probing, directly and in real time, coherent wavelength-tunable acoustic waves [13–15].

Due to the sandwich-like lamination and strong anisotropy of elastic properties caused by the NW-microstructure, the laser-generated wave fields are simulated within the elastodynamic model of anisotropic multilayered halfspace. The control of wave propagation in such structures presents a number of challenges. Indeed, their anisotropy complicates the physical phenomena associated with waves. Then, these waves have a multi-modal character (at a given wavelength or frequency several modes coexist) and they are dispersive (the phase and group velocities of each of the modes are frequency dependent). At present, the generally accepted means of numerical simulation are the finite element method (FEM) and similar mesh-based approaches. However, the FEM is not well suitable for the wave problems considered because it can provide the quantitative information only about the total wave field while the characteristics of individual SAW modes are of interest. Besides, the FEM can be too time-consuming due to the large relative lengthening of the simulated structures resulting in a sharp increase of the number of elements necessary for the correct SAW simulation. In view of the repeating recalculation of wave characteristics in the course of inverse problem solution (i.e., adjusting of the calculated SAW dispersion curves to the experimentally measured points), the efficiency of the simulation methods is of great importance.

To avoid all the difficulties discussed above, we use an analytically based computer model. It relies on the integral and asymptotic representations in terms of the elastodynamic Green's matrix of the laminated anisotropic half-space considered. An effective computer implementation is achieved by the use of fast and stable algorithms for the Green matrix, pole and residue calculations [16,17]. This computational model varies the materials properties (effective elastic constants, densities) to find a wave with such characteristics as were measured on experimental samples.

The rest of the paper is organized as follows. In Section 2 we describe the samples and experimental technique for exciting and detection of elastic waves. The computational model and its realization are described in Section 3, followed by test numerical results illustrating the dependence of GW propagation on material parameters in agreement with the experiment data. Finally, the discussion on effective elastic parameters is provided in Section 4 and Conclusions.

## 2. Materials and Experimental Set-Up

### 2.1. Samples

GaN nanowires were grown on a 2.5 nm-thin n-doped AlN buffer layer deposited on n-type doped Si(111) substrate by Plasma-Assisted Molecule Beam Epitaxy (PA-MBE) [18]. They were grown under nitrogen-rich conditions (V/III = 1.36) at temperature higher than 760 °C. The GaN NWs grown following this procedure are vertically oriented and present a hexagonal cross-section shape delimited by $\{10 - 10\}$ planes [19]. Depending on the growth conditions, the NW diameter ranges between 50 and 70 nm and their density evolves between $10^8$ and $10^{10}$ NW/cm$^2$. Finally, their length, which depends on the growth time, is approximately of $1.00 \pm 0.12$ µm.

The elastic properties of the GaN NW/HSQ composite layer, playing the role of active layer in piezoelectric devices, depend on the morphological properties of the NWs, which are embedded into a polymer matrix presenting lower elastic constants. In order to investigate the influence of NW characteristics, we have considered two types of NW morphologies. The first type of samples corresponds to GaN NWs grown at 800 °C (Figure 1). In these conditions, the NW density reaches $10^{10}$ NW/cm$^2$ and their diameter is around $50 \pm 10$ nm. The second type of samples has been grown at lower temperature (760 °C). Due to this growth temperature, a parasitic and thick 2D-layer is formed at the bottom of the NWs, as shown on Figure 2. These conditions lead to the formation of wider (diameter of $70 \pm 20$ nm) and less dense ($5 \times 10^8$ NW/cm$^2$) GaN NWs. This type of NWs will be referred to as "*pyramidal*" NWs.

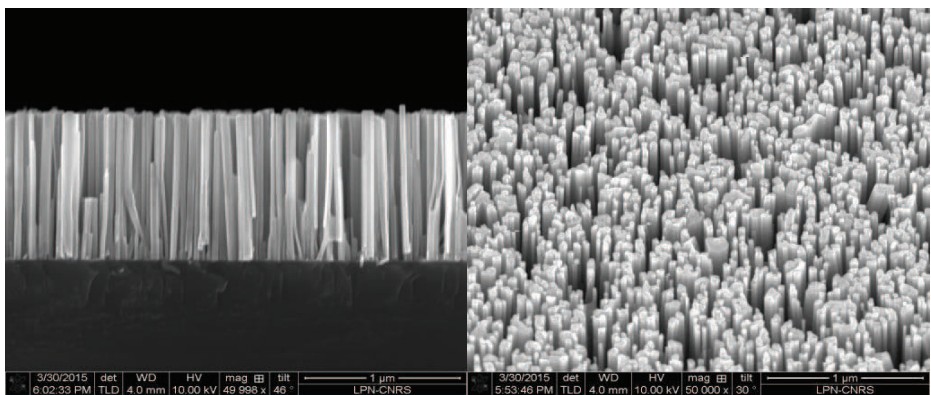

**Figure 1.** Scanning electronic microscope picture of GaN NWs (Gallium nitride nanowires) on n-doped Si (111) substrate before the encapsulation into a polymer matrix. All NWs are vertical.The density is of the order of $10^{10}$ NWs/cm$^2$. The filling factor is 33–40%.

To evaluate the effective elastic properties of GaN NWs-based composite structures, the as-grown GaN NWs of the two morphologies have been embedded into HSQ matrix. This matrix, spin-coated homogeneously onto the nanowire array, has been annealed at 400 °C to cure the polymer. Then reactive ion etching was used to remove polymer excess from the nanowire summits for metallic contact deposition. Finally, after a de-oxidation of the nanowire top part, CrPt contacts were deposited through a shadow mask with 2 mm openings and a bottom TiAu contact was made on the conducting silicon substrate. In addition, three samples corresponding to the second NW morphologies (i.e., with a rough parasitic layer) were prepared following the same procedure but with different annealing temperature (300 °C, 400 °C and 500 °C).

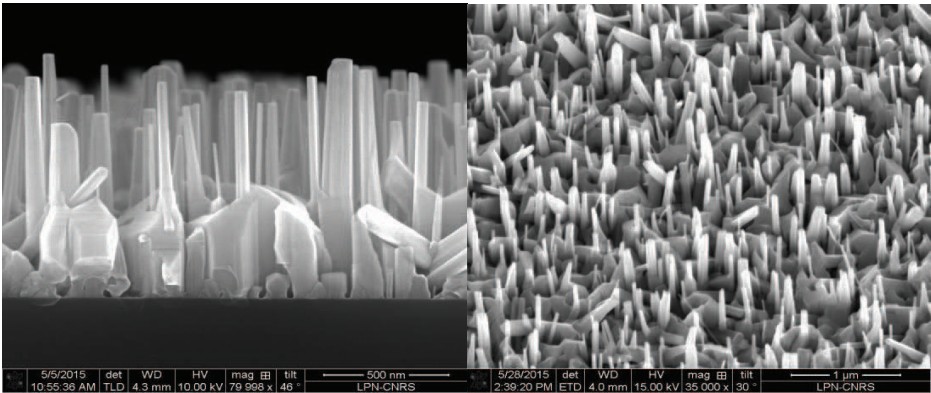

**Figure 2.** Scanning electronic microscope picture of GaN NWs on n-doped Si (111) substrate before the encapsulation into a polymer matrix. The NWs are growth at lower temperature than for sample (Figure 1), then leading to a smaller density and the appearance of a parasitic and thick-2D GaN layer located at the NW bottom here. The filling factor is less than 10%.

*2.2. Experimental Technique*

The properties of surface acoustic waves propagating in these composite structures were studied using a laser based transient grating (TG) method [13–15]. This technique is an optical method that allows to monitor surface acoustic waves stimulated in thin films with optical short pulses. It is useful for accurately and nondestructively characterizing the high frequency elastic properties.

In this approach, two laser pulses are spatially and temporally overlapped on the surface of the sample to excite elastic waves with a wave vector determined by the optical interference pattern generated by the crossed excitation pulses (Figure 3). These crossed infrared pulses (wavelength = 1064 nm; 30 ps in duration) heat the surface of samples, which creates thermoelastic stresses that finally relax in the form of two counter-propagating surface acoustic waves with wavelength $\Lambda$ equal to the interference fringes spacing and frequencies $f = \Omega/2\pi$. These acoustic waves are then monitored in real time by detecting the green light from a continuous-wave laser (wavelength = 532 nm) diffracted on the surface relief associated with the acoustic disturbances. The detection of elastic waves is made using a heterodyne scheme and fast detection electronics.

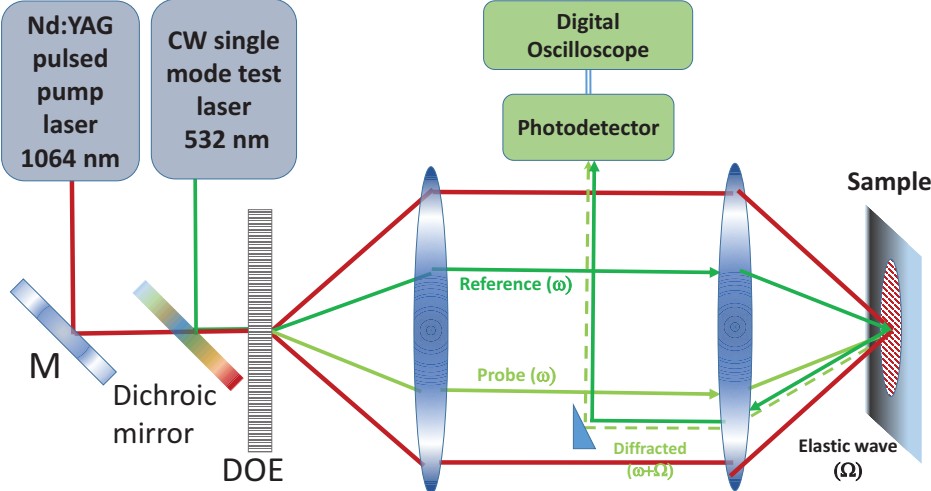

**Figure 3.** Optical set-up and laser system for TGM (transient grating method) experiment in reflection geometry. Two convergent lens; M: mirror; DOE: diffractive optical element.

Controlling the angle between the two incident pump beams via adjusting of the focal path allows to tune the wavelength $\Lambda$ of acoustic wave. The Fourier analysis of experimental data at fixed value

of $\Lambda$ shows distinct frequencies, each corresponding to distinct mode of the structure, as depicted in Figure 4a. Varing $\Lambda$ and recording the frequencies resonance peaks for each value, allows therefore to map the dispersion diagrams with the step size in wave vector determined by the pattern of diffractive optical element and the magnifying power of optical system.

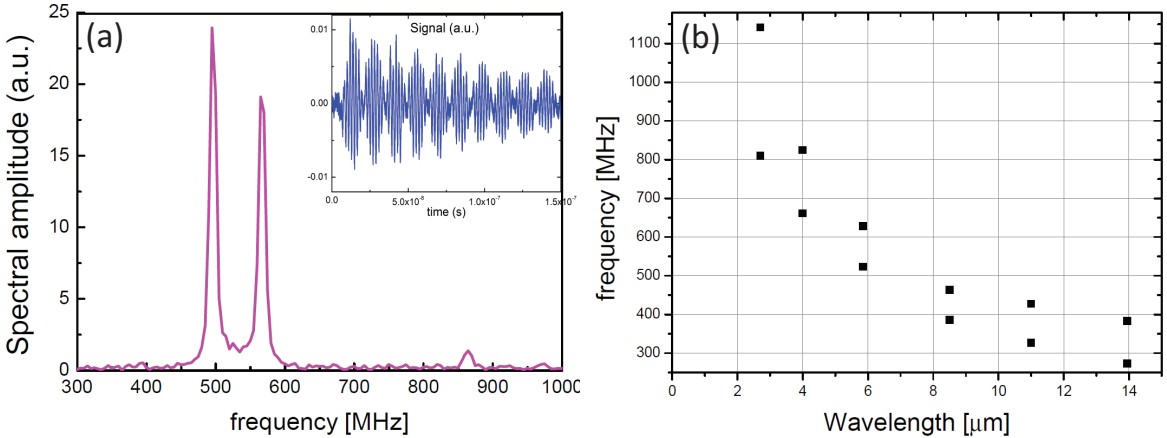

**Figure 4.** Typical frequency spectrum of the laser-generated signal shown in the insert TGM acquired for the SAW (surface acoustic wave) wavelength $\Lambda = 5.85\,\mu\text{m}$ (**a**). Typical dispersion diagram obtained in experiment (**b**).

## 3. Theoretical Framework

### 3.1. Premises

The theoretical model is based on the analysis of dispersion curves of guided surface acoustic waves. Resonance peaks in the frequency spectra of the acquired signals (e.g., Figure 4a) are dots on the wavelength-frequency plane $(\Lambda, f)$ (e.g., Figure 4b). They indicate that at the corresponding wavelength-frequency pairs $(\Lambda_j, f_j)$, the laser beam excites SAWs. To use this information for the probing (i.e., quantitative evaluation) of the effective material constants of HSQ-NW composite medium, one needs, first, a mathematical and computer model allowing adequate prediction of the SAW characteristics for a given set of input sample's parameters (geometry, elastic moduli, and density). Second, it is necessary to specify an objective function which determines the deviation of the calculated SAW dispersion characteristics from the experimentally obtained ones. The minimization of this function in the space of input parameters of the specimen should result in approaching of the calculated characteristics to the experimental data. Additionally, the set of variable parameters providing the global minimum can be treated as the effective sample's parameters within the framework of the model used.

The analysis of the scanning electronic microscope pictures of cleaved specimens (e.g., Figure 5a) tells us that, in general, the samples can be considered as three-layer elastic waveguides (Figure 5b). Counting top-down, the HSQ-NW composite is the core (second) layer covered by an electrical-contact coating (first layer) and underlain by a thick silicon substrate which may be simulated by an elastic half-space (third domain). It is expected that the total thickness of the first two layers deposited on the substrate is about 1 $\mu$m while the coating takes about one-tenth.

It is obvious that, due to a complex microstructure of the nanowired-based material, its elastic properties can be simulated neither by the properties of separate constituents (HSQ polymer or NW material) nor by their jointly averaged characteristics. Vertical elongation of the nano-inclusions results in a strong anisotropy of elastic properties that manifests itself in a large difference between vertical and horizontal deformations. Consequently, within the mathematical model, its stress-strain relations should be described by a transversely isotropic elastic medium with the horizontal plane of isotropy $(x, y)$ (Figure 5b). Such materials are specified by five independent elastic moduli $C_{ijkl}$ and density $\rho$.

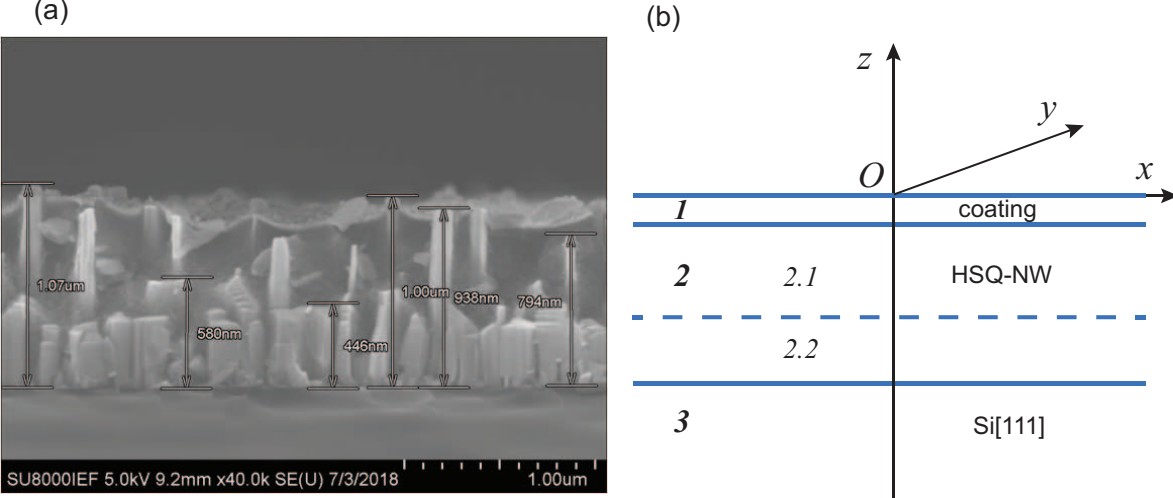

**Figure 5.** (**a**) Electronic microscope picture of the cross-section of the cleaved specimen. Here, NWs are embedded into HSQ matrix, Cr/Pt metal electrode is deposited on the top. (**b**) Coordinate system for a multilayered elastic half-space used to model SAWs; dashed line is for a further subdivision in accordance with vertical inhomogeneity of the HSQ-NW layer.

The material parameters of the CrPt alloy in the coating and Si[111] in the substrate are generally known and may be taken from literature. The underlying silicon half-space is of cubic symmetry specified by three independent elastic constants. The full set of Si[111] parameters used in the calculations is presented in Table 1.

**Table 1.** Unmodified input material constants for the coating CrPt and substrate Si[111]. Elastic moduli $C_{ij}$ [GPa], density $\rho$ [$10^3$ kg/m$^3$], thickness $h$ [nm], and the corresponding engineering constants: Young's moduli $E_x$, $E_z$ and shear modulus $\mu_z$ [GPa]; Poisson's ratios $v_x$ and $v_z$.

|  | $C_{11}$ | $C_{33}$ | $C_{13}$ | $C_{44}$ | $C_{66}$ | $\rho$ | $E_{xy}$ | $E_z$ | $\mu_z$ | $v_{xy}$ | $v_z$ |
|---|---|---|---|---|---|---|---|---|---|---|---|
| CrPt | 315 | 315 | 193 | 61 | 61 | 21.5 | | | | | |
| Si[111] | 195 | 205 | 44.7 | 60.7 | 70.3 | 2.33 | 175 | 189 | 60.7 | 0.241 | 0.179 |

The first layer (coating) is isotropic (two independent elastic constants that are expected to be known, in such cases the values from Table 1 are used as input). However, due to the roughness of the top surface, the averaged elastic moduli of the top layer tend to be smaller than those of the pure CrPt alloy. Therefore, the effective material constants of the first layer and its thickness $h_1$ are also generally unknown and, in many cases, it is determined simultaneously with the HSQ-NW parameters.

The engineering constants are related to the elastic moduli through the connection between the $6 \times 6$ stiffness matrix $C$, which elements $C_{ij}$ are written in the Voigt notation, and the compliance matrix $S$: $C = S^{-1}$. In the case of transversely isotropic medium with the vertical axis of symmetry $Oz$, the compliance matrix $S$ written in terms of the Young modulus $E_x = E_y$ and $E_z$, shear modulus $\mu_z$, and Poisson's ratios $v_x = v_y$ and $v_z$ takes the form

$$S = \begin{pmatrix} S_1 & 0 \\ 0 & S_2 \end{pmatrix}$$

$$S_1 = \begin{pmatrix} 1/E_x & -v_x/E_x & -v_z/E_z \\ -v_x/E_x & 1/E_x & -v_z/E_z \\ -v_z/E_z & -v_z/E_z & 1/E_z \end{pmatrix}$$

$$S_2 = \mathrm{diag}(1/\mu_z, 1/\mu_z, 1/\mu_x), \quad \mu_x = E_x/(2(1+v_x))$$

Accordingly,

$$C = \begin{pmatrix} C_1 & 0 \\ 0 & C_2 \end{pmatrix}, \quad C_1 = S_1^{-1}, \quad C_2 = S_2^{-1}$$

A three-layer elastic anisotropic half-space is a basic model for the SAW simulation. However, since some samples present straight and homogeneously grown NWs (e.g., Figure 1), in other samples, the NWs are of "*pyramidal*" shape (Figure 2) in the sense where the straight NWs present a large base resulting from the parasitic layer grown at their bottom due to the lower temperature. It means that the elastic properties and density of the HSQ-NW layer vary with the depth, and it should be modeled by a vertically inhomogeneous (layered or functionally gradient) elastic medium. Nevertheless, at the first stage, such a core layer can also be roughly modeled by a homogeneous transversely isotropic material. And the numerical analysis shows that such an approximation provides quite reasonable results. However, judging by the pictures of "*pyramidal*" NWs, it is more precise to divide such a layer in two sublayers with different material properties (e.g., sublayers 2.1 and 2.2 separated in Figure 5b by the horizontal dashed line).

Thus, the wave processes in the samples under study are simulated on the basis of solutions to the elastodynamic boundary value problems (BVP) for multilayered anisotropic half-spaces.

### 3.2. Simulation

To simulate the frequency spectra of the laser-generated SAWs, the BVP is formulated with respect to the complex displacement amplitude $\mathbf{u} = (u_1, u_2, u_3)$ of the time-harmonic oscillation $\mathbf{u}(\mathbf{x})e^{-i\Omega t}$; $\Omega = 2\pi f$ is angular frequency, $f$ is frequency, $\mathbf{x} = (x, y, z)$ is a point in the Cartesian coordinate system; plane $(x, y)$ and axis $z$ are aligned with the sample's surface and outward normal (Figure 5b).

The components of the displacement vector $\mathbf{u}$ obey the elastodynamics equations

$$C_{ijkl}u_{l,jk} + \rho\Omega^2 u_i = 0, \quad i = 1, 2, 3. \tag{1}$$

The elastic stiffness tensor $C_{ijkl}$ and the density $\rho$ are piecewise constant functions of the transverse coordinate $z$, thereby keeping constant values within the sublayers of thickness $h_m$, $m = 1, ..., M$. The bottom substrate is a half-space ($h_M = \infty$). The outer surface $z = 0$ is stress free, except within the loading region $D$:

$$\boldsymbol{\tau}|_{z=0} = \mathbf{q}, \quad (\mathbf{q}(x, y) \equiv 0 \text{ for } (x, y) \notin D), \tag{2}$$

and the sublayers are perfectly bonded with each other. Here $\boldsymbol{\tau} = \{\tau_{xz}, \tau_{yz}, \tau_z\}$ is a tension vector at a horizontal surface area $z = $ const, $\mathbf{q}$ is a given load simulating the action of the laser beam generating SAWs.

The solution to this BVP is obtained in terms of the Green's matrix $k(\mathbf{x})$ and the vector of load $\mathbf{q}$ applied to a surface area $D$, or equivalently, via their Fourier symbols $K(\alpha, \alpha_2, z) = \mathcal{F}_{xy}[k]$ and $\mathbf{Q}(\alpha_1, \alpha_2) = \mathcal{F}_{xy}[\mathbf{q}]$ [16,17,20]:

$$\begin{aligned}
\mathbf{u}(\mathbf{x}) &= \int\int_D k(x - \xi, y - \eta, z)\mathbf{q}(\xi, \eta)d\xi d\eta \\
&= \frac{1}{(2\pi)^2} \int_{\Gamma_1}\int_{\Gamma_2} K(\alpha_1, \alpha_2, z)Q(\alpha_1, \alpha_2)e^{-i(\alpha_1 x + \alpha_2 y)}d\alpha_1 d\alpha_2.
\end{aligned} \tag{3}$$

Here $\mathcal{F}_{xy}$ is the Fourier transform with respect to $x$ and $y$ variables; the Fourier parameters $\alpha_1$ and $\alpha_2$ play the role of wavenumbers for the waves propagating along the $(x, y)$ surface. The integration paths $\Gamma_1$ and $\Gamma_2$ go along the real axes, deviating from them into the complex planes $\alpha_1$ and $\alpha_2$ for rounding the real poles $\zeta_n$ of the matrix $K$ elements.

The frequency domain Green's matrix $k = (\mathbf{k}_1 \vdots \mathbf{k}_2 \vdots \mathbf{k}_3)$ is formed from the solution vectors $\mathbf{k}_j$ corresponding to the point loads applied along the basic coordinate vectors $\mathbf{i}_j$. In the case under consideration, $\mathbf{k}_j = \mathbf{u}$, where $\mathbf{u}$ is the solution to BVP (1) and (2) with $\mathbf{q} = \mathbf{i}_j \delta(x, y)$ in the boundary conditions at the surface $z = 0$; $\delta(x, y)$ is Dirac's delta-function. The exhaustive description of the algorithms of matrix $K$ calculation can be found in Refs. [16,20]. Ibid, as well as in Reference [17], the derivation of asymptotic representations for SAWs generated by the surface load $\mathbf{q}$ is described in details.

The SAWs are extracted from the path integrals of Equation (3) as the residues from the real and nearly real poles $\zeta_n$. Since the action of the laser beam on the sample's surface can be simulated by a vertical load $\mathbf{q} = (0, 0, q)$, only the third column of matrix $K$ is involved in the representation of the laser-generated wave field $\mathbf{u}$. Moreover, since only the third (vertical) component $u_3 = u_z$ of the excited displacement $\mathbf{u}$ is laser acquired as the sample's response, only one element $K_{33}$ of the matrix $K$ is needed to simulate the experimental measurements. Under these conditions, the SAW asymptotics of Reference [16] is reduced to the form

$$u_z(r) = \sum_{n=1}^{N} a_n e^{i\zeta_n r} / \sqrt{\zeta r} + O((\zeta r)^{-1}), \quad \zeta r \to \infty, \ r\sqrt{x^2 + y^2}$$

$$a_n = \sqrt{\zeta / (2\pi i \zeta_n)} \operatorname{res}K_{33}(\alpha_1, 0, 0)|_{\alpha_1 = \zeta_n} Q(\zeta_n, 0)$$

(4)

Here $N$ is the number of real and closest to the real axis complex poles $\zeta_n$ held in the asymptotics, $\zeta$ is a characteristic wavenumber; for definiteness, the amplitude factors are shown for the SAWs propagating in the $x$-direction ($y = 0$, $\alpha_2 = 0$, $r = x$). Each term of expansion (4) describes the guided wave, for which the pole $\zeta_n$ is the wavenumber. With real $\zeta_n$, they are traveling waves propagating with the phase and group velocities $c_n = \Omega / \zeta_n$ and $v_n = d\Omega / d\zeta_n$ or slownesses $s_n = \zeta_n / \Omega$. The imaginary part $\operatorname{Im}\zeta_n$ of a complex $\zeta_n$ results in the exponential attenuation $O(e^{-|\operatorname{Im}\zeta_n|r})$ as $r \to \infty$, and the corresponding waves are leaky or pseudo-surface acoustic waves (PSAW).

The poles $\zeta_n$ are the roots of the $K_{33}$ denominator, therefore, the characteristic equation that relates the wavelength $\Lambda = 2\pi / \alpha_1$ with frequency $f$ can be written in the form

$$K_{33}(\Lambda, f) = 0,$$

(5)

where $K_{33}$ is treated as a function of $\Lambda$ and $f$ at $\alpha_2 = 0$ and $z = 0$. With a fixed wavelength $\Lambda$, the roots of this equation $f_n$ are the frequencies of SAW/PSAW modes, which should coincide with the TGM measured frequencies of the sample's resonance response, e.g., peak frequencies in Figure 4a. And vice versa, with a fixed frequency $f$, the roots $\Lambda = \Lambda_n(f)$ yield the SAW wavenumbers $\zeta_n = 2\pi / \Lambda_n$.

### 3.3. Validation

The magnitude of $|K_{33}(\Lambda, f)|$ reaches the maximal values at the sets $(\Lambda, f_n(\Lambda))$ specified by the roots of characteristic equation (5); theoretically it is infinite with real $\zeta_n$. Therefore, the ridges in the level-line plots depicting $|K_{33}|$ as a function of $\Lambda$ and $f$ visually show the SAW/PSAW dispersion curves in the plane $(\Lambda, f)$. To confirm the ability of the developed Green's matrix based model to correctly simulate SAWs generated in the samples, we had to make sure that the ridges pass through the experimentally obtained points $(\Lambda_j, f_j)$ indicating the peaks of the frequency spectrum for a sample with known material parameters.

First of all, such a validating comparison has been performed for a simple two-layer specimen Ni/Si. The material properties of the isotropic nickel coating were taken from the handbooks ($C_{11} = 247$ GPa, $C_{44} = 122$ GPa, and $\rho = 8900$ kg/m$^3$) while the silicon substrate is the same in all experiments (Table 1). In this sample, the thickness of the coating ($h_1 = 15$ nm) was very small as compared with the range of SAW wavelengths (2–15 μm) in the measurements. Therefore, its influence was insignificant and the only Rayleigh-like SAW was excited, and, as expected, the only ridge of the

calculated surface $|K_{33}(\Lambda, f)|$ passed through the experimental points (Figure 6). This has confirmed both the measurement and simulation accuracy.

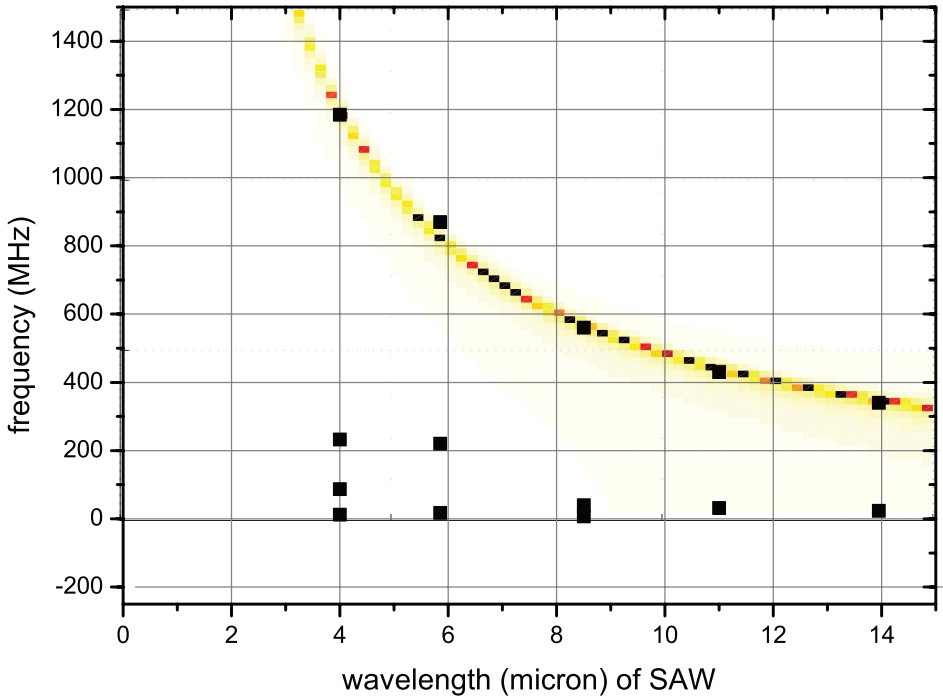

**Figure 6.** Validating example for the test sample Ni/Si. Calculated within the model developed dispersion curve of the only Rayleigh-like SAW indicated by the ridge in the $|K_{33}|$ level-line image (yellow) passes through the TGM measured points (markers); low-frequency markers ($f < 250$ MHz) are just hardware interference.

Figure 7 presents the results of the theory-to-experiment comparisons with a more complex three-layer specimen Ni/HSQ400/Si. The core layer is a pure HSQ polymer, still without NWs, annealed at the temperature $t = 400\,^\circ$C. This material is isotropic, therefore, it was not too difficult to determine its material constants manually. It required just a few turns to fit the $|K_{33}|$ ridges (Figure 7b) to the experimental points $(\Lambda_j, f_j)$ (Figure 7a). The fit is achieved with the HSQ parameters $C_{11} = 11.89$ GPa, $C_{44} = 2.97$ GPa, and $\rho = 1000$ kg/m$^3$.

The real roots of Equation (5) calculated with these parameters allowed tracing the dispersion curves $f = f_n(\Lambda)$ (Figure 7c). Obviously, the curves go along the $|K_{33}|$ ridges but not the whole length. For example, the two upper-right experimental points get on the ridge in Figure 7b, but in Figure 7c this area is blank. The reason is that this part of the ridge is already associated with the complex pole $\zeta_n$, i.e., the corresponding SAW actually becomes PSAW (leaky wave). The transformation of a traveling surface wave into a leaky wave occurs when its phase velocity $c_n$ becomes greater than the velocity $v_s$ of the bulk shear waves in the bottom half-space (i.e., in the silicon substrate in the case). Or equivalently, when its slowness $s_n = 1/c_n$ is less then the substrate's S-wave slowness $S3 = 1/v_s$. Depicting dispersion curves in the frequency-slowness plane is more convenient, since, in contrast with the phase velocity curves that can come down from infinity, the slowness magnitudes vary in a limited range specified by the frequency independent bulk-wave slownesses. In Figure 7d and similar figures below, the latter are shown by horizontal dashed lines. They are indicated by the symbols $P1$, $P2$, $P3$, and $S1$, $S2$, $S3$, which relate to longitudinal and transversal (*P* and *S*) bulk waves propagating in the $x$-direction in each of the sublayers numbered from top to bottom. One can see that those two points, getting in the blank area in Figure 7c, lie below the $S3$ line in Figure 7d.

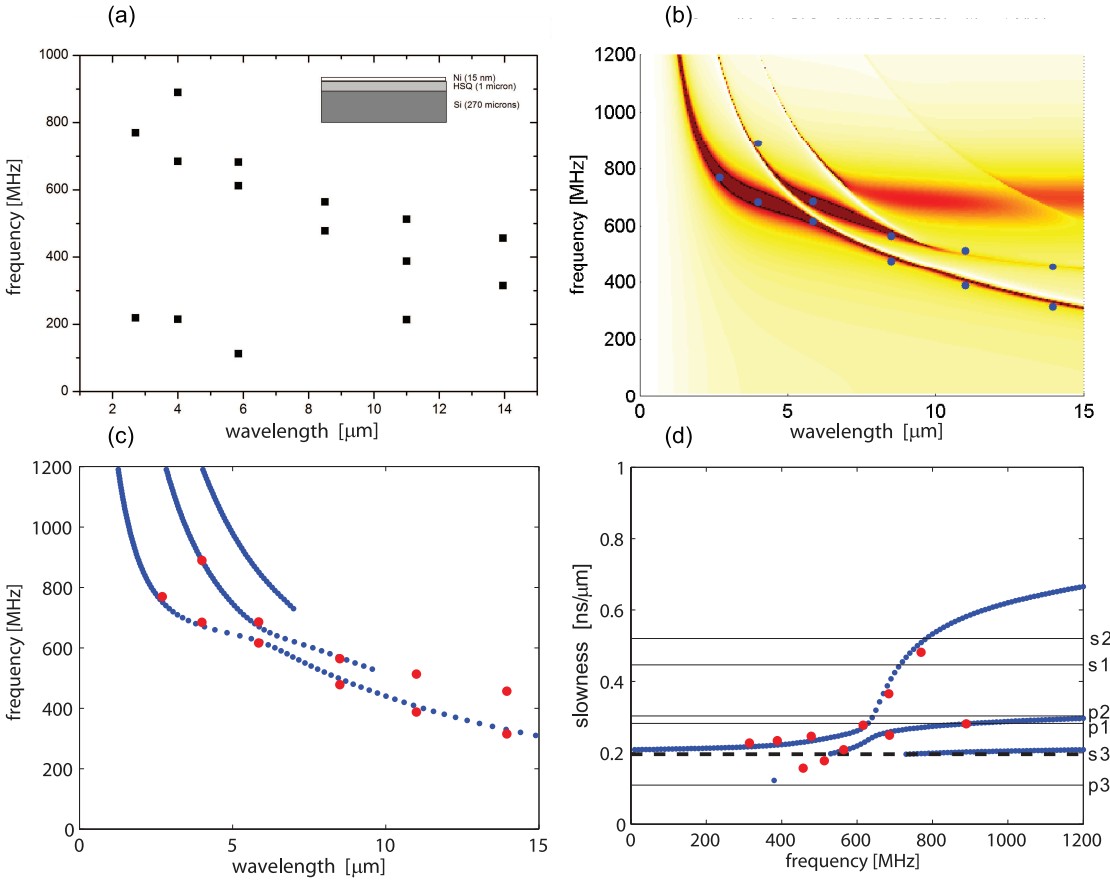

**Figure 7.** Second validating example for the sample Ni/HSQ400/Si without NWs. Experimental points in the $(\Lambda, f)$ plane (**a**); $|K_{33}(\Lambda, f)|$ leve-line image (**b**); SAW dispersion curves and experimental points (bold dots) in the $(\Lambda, f)$ and frequency-slowness planes (**c**,**d**); low-frequency markers ($f < 250$ MHz) in part (**a**) are hardware interference.

In general, Figure 7 demonstrates a good match of the SAW dispersion curves calculated for manually selected HSQ parameters with the experimental data. However, the HSQ-NW composite, which are the main goal, do not promise such an easy selection of proper effective parameters. Creating an algorithm of the fitting, i.e., the development of methods for the inverse problem solution, is necessary.

### 3.4. Inverse Problem

To obtain the effective elastic moduli of the interlayer HSQ-NW, the sample parameters (the matrix of elastic moduli $C$, the density $\rho$ and the thickness $h$ of each sublayer), which are inputs to calculate $K$ within the multilayered mathematical model used, vary to match the experimental points. This goal is achieved through the minimization of a certain objective function that can be constructed in various forms. A natural way is to minimize the discrepancy $\Delta$ between the measured ($d_j^m$) and calculated ($d_j^c$) SAW dispersion characteristics (e.g., between the SAW wavelengths or group velocities at the measured resonance frequencies $f_j$):

$$\Delta(C, \rho, h) = \sum_j (1 - d_j^c / d_j^m)^2 \tag{6}$$

To search for the global minimum of such objective functions, genetic algorithms are usually employed [21]. For example, a similar approach was successfully applied to the evaluation of the effective elastic properties of layered composite fiber-reinforced plastic plates [22].

However, the calculation of dispersion characteristics at each step is unreasonably time-consuming, because, to obtain each of them, the matrix $K$ are to be computed hundreds or even thousands of times. Drastic cost reduction is achieved by using the objective function

$$F(C, \rho, h) = \sum_j |K_{33}^{-1}(\Lambda_j, f_j)|, \tag{7}$$

where $(\Lambda_j, f_j)$ are experimentally obtained points in the wavelength-frequency plane. The calculation of this function requires only one call of the procedure $K_{33}$ for each experimental point. The appropriateness of function (7) as an objective function is explained by the fact that it becomes equal to zero when the variable set of sample parameters $C$, $\rho$, and $h$ yields, together with the fixed known parameters, the SAW characteristics $d_j^c$ corresponding to the measured resonance points $(\Lambda_j, f_j)$. A similar approach was also recently proposed for real-time assessment of anisotropic plate properties using elastic guided waves [23]. To obtain the set of parameters providing the minimal value of function $F(C, \rho, h)$ (i.e., to find the effective sample's parameters), we use the method of coordinate-wise minimization within prescribed ranges of parameter variations. It usually requires no more than 50 iterations, which is computationally cost free even as compared with the plotting of the dispersion curves obtained.

## 4. Results and Discussion

### 4.1. Meaning of Effective Parameters

The method developed for determining effective parameters of nanocomposite waveguide structures has been tested with a number of samples differing in NW morphology and annealing temperature. In this section, we present some of these results (dispersion curves and tables of effective parameters) illustrating the influence of these two factors on the effective constants of the HSQ-NW composites and the waveguide properties of the related sandwich structures.

It is worth to note that the obtained sets of parameters should not be treated as the "real" parameters of the corresponding materials. First, because such "real" parameters do not exist in the nature at all, while the material constants introduced within one or other model of a solid medium are just a tool to simulate its stress-strain state. Second, the effective parameters are obtained to simulate wave phenomena in a layered structure as a whole. Therefore, a change of properties of one lamina (e.g., its thickness) affects the properties of all others. Moreover, the borders between the layers, in fact, also do not exist but are introduced manually or obtained by minimizing the goal function. For example, the density of some layers does not correspond to these real mass divided by the volume but is just a constant in governing Equation (1) providing the required SAW characteristics. The same reasoning is valid for all other effective parameters.

On the other hand, if a specimen were fabricated strictly from $M = 3$ or other number of homogeneous layers used in the mathematical model with the elastic properties described by the effective constants obtained, it would exhibit the same wave properties as the sample under study. And most importantly, since the SAW eigenfrequencies and wavenumbers calculated within the mathematical model are the same as those measured in the experiments, their eigenforms should be very similar to those in the real samples. The eigenforms determine the laws of deformation in the sample at the corresponding frequencies, thus, its electrical output due to the deformation of the embedded piezoelectrical NWs. This can be used for the design and optimization of piezogenerators based on III-nitride NWs.

As for the mechanical properties of the microstructural material HSQ-NW itself, the narrower ranges of the parameters of other layers are specified, the more stable its effective moduli are obtained. Therefore, to analyze the tendencies of their change, e.g., depending on the annealing temperature, the best is to fix the properties of the coating and substrate as well as all thicknesses even if they do not provide the global minimum in the space of all input parameters of the model. The results of

such estimation of the temperature influence obtained with fixed layer thicknesses are discussed in Section 4.4 below.

### 4.2. High Density Straight NWs and Rare NWs with Thick 2D-Layer

First numerical experiments for nanowires-in-polymer samples have been carried out for the two samples shown on Figures 1 and 2. The first sample presents straight NWs, relatively homogeneously distributed in plane and embedded in the HSQ polymer. The second has rare NWs, very dispersive in form and in sizes ("*pyramidal*" NW) . Table 2 displays the effective parameters of the coating and core layers of these samples obtained by minimizing cost function (7) constructed on the basis of three-layer model ($M = 3$). Figures 8 and 9 illustrate the dispersion properties of these samples simulated with these parameters. The left parts (a) of these figures are level-line images of the $|K_{33}|$ surface, and the right parts (b) are SAW dispersion curves in the frequency-slowness plane. The experimental results are shown on the both parts by bold markers. One can see that in the case of nanowired interlayers, the theoretical plots are also in good agreement with them.

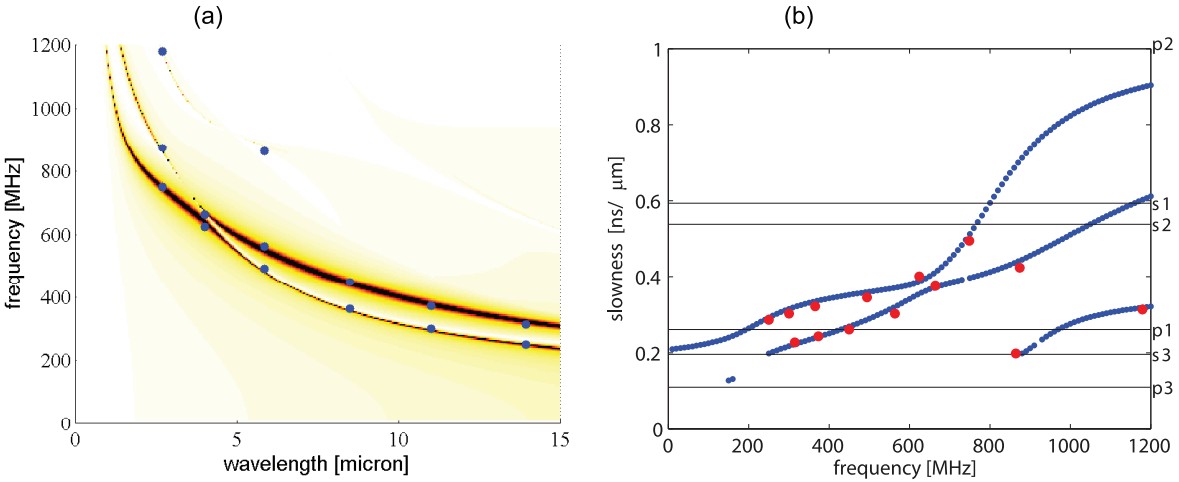

**Figure 8.** Experimental points (bold dots), $|K_{33}|$ level-line image (**a**) and slowness dispersion curves (**b**); straight-NW sample Figure 1.

**Table 2.** Effective parameters of the HSQ-NW composition obtained for the samples Figure 1 (straight NWs, $t = 400\,^\circ$C.) and Figure 2 (rare NWs and thick 2D-layer, $t = 400\,^\circ$C.).

|  | $C_{11}$ | $C_{33}$ | $C_{13}$ | $C_{44}$ | $C_{66}$ | $\rho$ | h | $E_x$ | $E_z$ | $\mu_z$ | $\nu_x$ | $\nu_z$ |
|---|---|---|---|---|---|---|---|---|---|---|---|---|
| sample Figure 1 | 2.00 | 129 | 1.26 | 7.12 | 0.74 | 2.06 | 1059 | 1.85 | 128 | 7.12 | 0.26 | 0.499 |
| sample Figure 2 | 20.8 | 106 | 0.00 | 10.4 | 5.35 | 1.77 | 887 | 15.9 | 106 | 10.4 | 0.485 | 0.00 |

As was expected, the effective moduli of the HSQ-NW layers of these samples exhibit large difference for $x$ and $z$ directions (e.g., there is a strong contrast between Young moduli $E_x$ and $E_z$ in Table 2) confirming a strong anisotropy of this new composite material. If we take a basic isotropic approximation of bulk values for the two constituting materials weighted by the filling factor (33–40% for Figure 1 and 6–8% Figure 2), the estimated in-plane and out-of-plane Young moduli of our HSQ-GaN NW composite layer are in the range 103–123 GPa and 26–32 GPa respectively. Comparing these values with the restored values of Table I for the first morphology (Figure 1), we observe that $E_z$ is close to this basic estimation, whereas the $E_{xy}$ is one order of magnitude different.

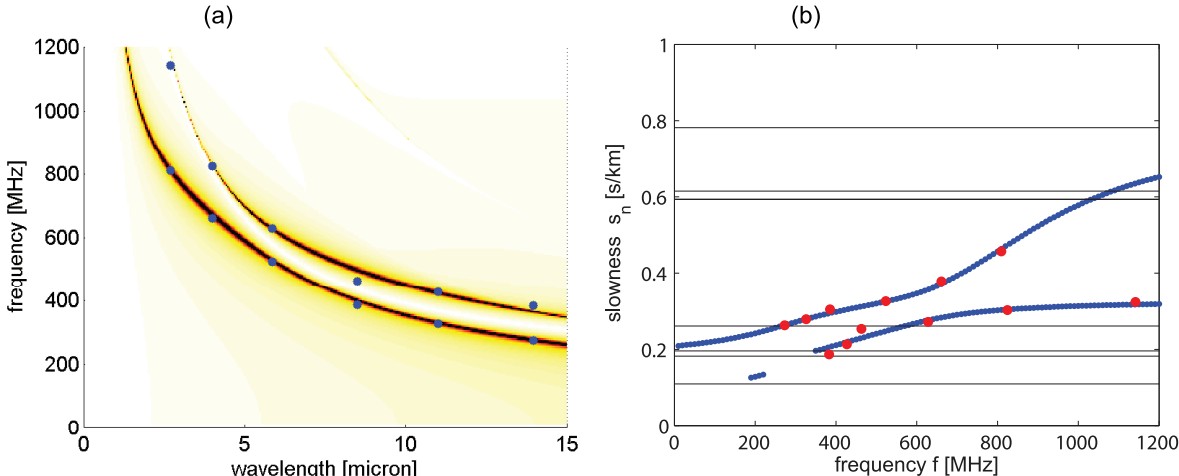

**Figure 9.** Experimental points (bold dots), $|K_{33}|$ level-line image (**a**) and slowness dispersion curves (**b**); "*pyramidal*" sample Figure 2.

Concerning the second morphology (Figure 2), the both values are far from the estimated with bulk moduli. This discrepancy may arise from the use of three layers model. Indeed, the parasitic rough layer changes considerably the elastic properties of the lower part of the composite medium. The simulation can be refined by dividing the composite medium into two sublayers.

### 4.3. Effect of Number of Sublayers

As was noted above, vertical inhomogeneity in the HSQ-NW composites with "*pyramidal*" NWs assumes subdivision of the core layer into sublayers (Figure 5). To estimate the effect of such finer simulation, the effective parameters of "*pyramidal*" shape NWs have been determined within the three-layer and four-layer models for one example (Table 3, Figure 10).

One can see that, though the elastic constants and density are sufficiently different with $M = 3$ and $M = 4$, both models ensure the passage of the dispersion curves through the experimental points. The general form of the curves is also little affected by the transition from $M = 3$ to $M = 4$. The difference in the HSQ-NW effective parameters can be explained by the notable difference in the obtained thicknesses, which were also variable: $h_2 = 1001$ nm with $M = 3$ and $h_2 = 314 + 528 = 824$ nm with $M = 4$.

**Table 3.** Effective parameters of the CrPt coating and HSQ-NW400 composition obtained for the sample shown in Figure 2 within the 3-layer ($M = 3$) and 4-layer ($M = 4$) models ("*pyramidal*" NWs, $t = 400\,^\circ$C)

|          | $C_{11}$ | $C_{33}$ | $C_{13}$ | $C_{44}$ | $C_{66}$ | $\rho$ | h    | $E_x$ | $E_z$ | $\mu_z$ | $\nu_x$ | $\nu_z$ |
|----------|----------|----------|----------|----------|----------|--------|------|-------|-------|---------|---------|---------|
| CrPt     |          |          |          |          |          |        |      |       |       |         |         |         |
| $M = 3$  | 349      | 349      | 210      | 70       | 70       | 20     | 80   | 3.21  | 47.0  | 8.14    | 0.48    | 0.24    |
| $M = 4$  | 344      | 344      | 222      | 61       | 61       | 19     | 144  |       |       |         |         |         |
| HSQ-NW400 |         |          |          |          |          |        |      |       |       |         |         |         |
| $M = 3$  | 4.21     | 47.7     | 1.50     | 8.14     | 1.09     | 1.52   | 1001 | 3.21  | 47.0  | 8.14    | 0.48    | 0.24    |
| $M = 4$, 2.1 | 50.2 | 80.8     | 6.55     | 2.92     | 17.5     | 2.92   | 314  | 45.4  | 79.5  | 2.92    | 0.30    | 0.10    |
| 2.2      | 80.5     | 94.9     | 14.2     | 363      | 31.4     | 3.13   | 528  | 75.3  | 90.8  | 363     | 0.20    | 0.15    |

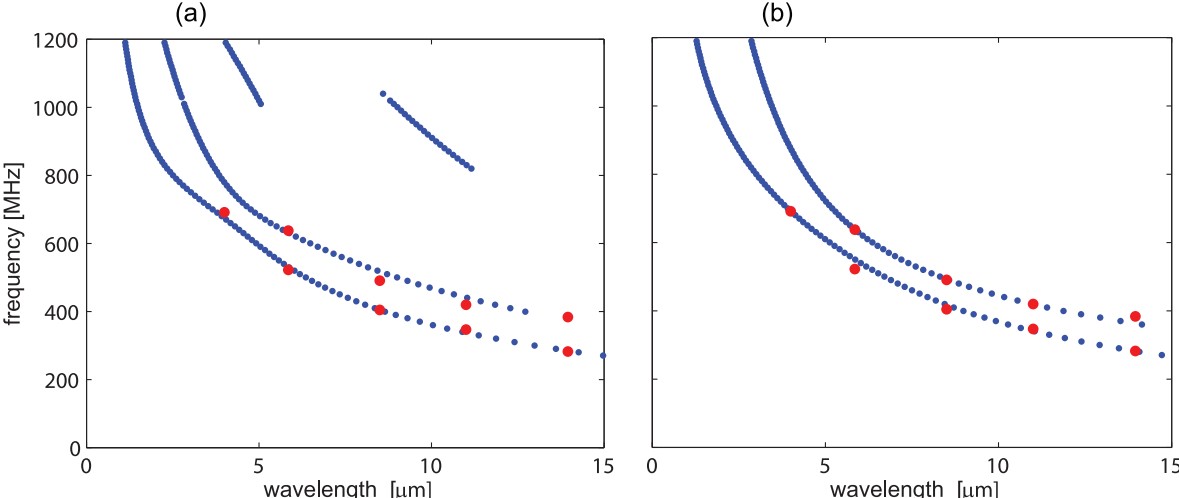

**Figure 10.** Dispersion curves calculated for the sample shown in Figure 2 within (**a**) $M = 3$ and (**b**) $M = 4$ models with the effective parameters from Table 3.

### 4.4. Influence of Annealing Temperature

To estimate the influence of annealing temperature on the HSQ-NW mechanical properties, a set of samples with "*pyramidal*" NW morphology similar to Figure 5 and different annealing temperatures (300 °C, 400 °C and 500 °C) has been fabricated and experimentally studied. These three samples have been grown at different growth runs, therefore they exhibit some variation of NW density and of the rough layer. To assure a comparability of the HSQ-NW parameters, we have prescribed the thicknesses of the sublayers based on the scan in Figure 5a: $h_1 = 150$ nm, $h_{2,1} = 350$ nm, $h_{2,2} = 550$ nm ($M = 4$). The first layer is thicker than the CrPt coating itself ($\approx$25–30 nm). It includes a zone of surface roughness and a bit of a HSQ-NW top part. Therefore, in the course of minimization, its parameters were also variable except the prescribed $h_1 = 150$ nm. This explains why the obtained Young moduli of the coating layer are different in different samples.

The results are summarised in Table 4; the corresponding slowness dispersion curves and the experimental points are shown in Figure 11.

**Table 4.** Effective parameters of the samples with the same NW "*pyramidal*" morphology (Figure 5a) fabricated with different annealing temperatures $t$, 4-layer model, fixed $h_1 = 150$, $h_2 = 350$, and $h_3 = 550$ [nm].

| | $C_{11}$ | $C_{33}$ | $C_{13}$ | $C_{44}$ | $C_{66}$ | $\rho$ | h | $E_x$ | $E_z$ | $\mu_z$ | $\nu_x$ | $\nu_z$ |
|---|---|---|---|---|---|---|---|---|---|---|---|---|
| $t = 300\,°C$ | | | | | | | | | | | | |
| CrPt | 490 | 490 | 366 | 61.7 | 61.7 | 20.6 | 150 | 176 | 176 | 61.7 | 0.48 | 0.48 |
| 2.1 | 54.4 | 70.6 | 14.9 | 3.05 | 17.0 | 3.25 | 350 | 45.5 | 64.7 | 3.05 | 0.34 | 0.20 |
| 2.2 | 39.2 | 90.9 | 4.14 | 150 | 19.1 | 3.05 | 550 | 39.0 | 90.0 | 150 | 0.02 | 0.10 |
| $t = 400\,°C$ | | | | | | | | | | | | |
| CrPt | 357 | 357 | 230 | 63.7 | 63.7 | 19.0 | 150 | 177 | 177 | 63.7 | 0.39 | 0.39 |
| 2.1 | 47.7 | 80.8 | 5.94 | 3.18 | 18.9 | 3.21 | 350 | 45.4 | 79.5 | 3.18 | 0.20 | 0.10 |
| 2.2 | 97.1 | 111 | 16.8 | 442 | 39.3 | 2.77 | 550 | 91.9 | 106 | 442 | 0.17 | 0.15 |
| $t = 500\,°C$ | | | | | | | | | | | | |
| CrPt | 335 | 335 | 184 | 75.1 | 75.1 | 13.9 | 150 | 204 | 204 | 75.1 | 0.36 | 0.36 |
| 2.1 | 50.1 | 56.3 | 29.3 | 17.3 | 11.1 | 2.88 | 350 | 30.1 | 33.3 | 17.3 | 0.36 | 0.38 |
| 2.2 | 56.2 | 73.7 | 6.66 | 1504 | 27.5 | 1.48 | 550 | 55.7 | 72.1 | 1504 | 0.01 | 0.12 |

It is known that the material hardness of HSQ polymer increases with the annealing temperature [24]. Therefore, one would expect to observe an increase of Young moduli between samples annealed at 300 °C, 400 °C and 500 °C. However, the obtained dependence of the effective

HSQ-NW constants is not monotone. We attribute this discrepancy to the variation of the NW morphology between the three samples. Indeed, the Young modulus on GaN is much lager than the one on HSQ. Therefore, a small variation of the NW density produces a stronger effect than the variation of the HSQ parameters with annealing. Nevertheless, it is seen in Figure 11 that the dispersion curves smoothly transform with the annealing temperature, which allows to trace the variation of the behavior of guided waves in such composite structures.

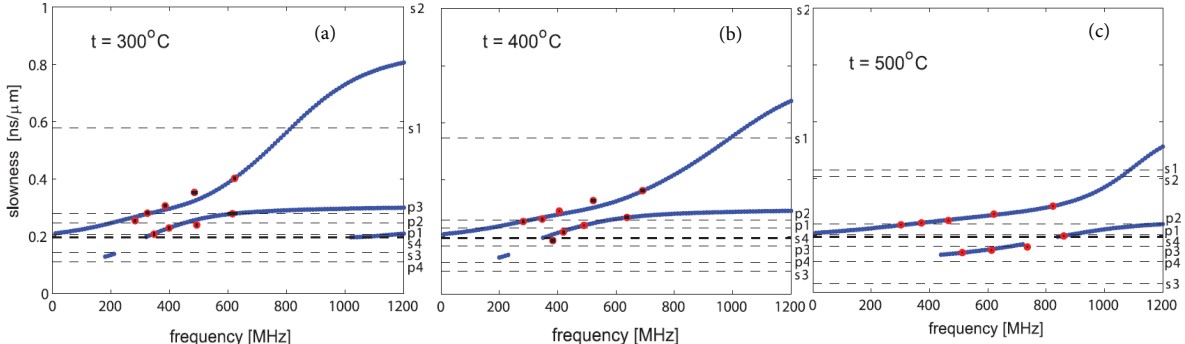

**Figure 11.** Illustration of the influence of annealing temperature; "*pyramidal*" NWs as in Figure 5a at different temperatures: (**a**) $t$ = 300 °C, (**b**) $t$ = 400 °C and (**c**) $t$ = 500 °C; effective input parameters are in Table 4.

## 5. Conclusions

In this work we took advantage of the highly dispersive behavior of an elastic wave guided on the surface of a substrate coated with a thin layer to extract the elastic parameters of the latter. The dispersion curves are sensitive to each of the elastic constants of constituent materials, to the filling factor of NW in the polymer matrix, and to the morphology of these inclusions. The Green's matrix formalism, that we have described in details, allows the retrieval of the effective elastic parameters of each sublayer in the stack. For all the samples we have tested, and whatever the annealing temperature is, excellent agreement is obtained between the simulation and the experimental data, both for the slowness values and for the relative intensity of the modes (i.e., the amplitude of $K_{33}$ component), allowing for a reliable determination of the elastic constants of the GaN NW's carpet. This approach that takes account of the natural spatial variation of the elastic constants caused by the "*pyramidal*" shape of NWs, allows provision of a set of elastic parameters fully characterizing a sample.

**Author Contributions:** L.L., E.C., O.B. and B.B. carried out the experiment. O.B. wrote the manuscript with support from E.G., N.G., B.B. and M.T., E.G. and N.G. developed the theoretical formalism, performed the analytic calculations and performed the numerical simulations. L.L., N.G., M.T. and F.J. fabricated the samples.

**Funding:** The experimental part of the work was supported by the by EU Horizon 2020 ERC project 'NanoHarvest' (Grant 639052), the French National Research Agency though the GANEX program (ANR-11-LABX-0014), and the theoretical part is supported by the Russian Science Foundation (Project No. 17-11-01191).

**Acknowledgments:** The authors would like to thank Emmanuel Peronne for helpful discussions.

**Conflicts of Interest:** The authors declare no conflict of interest.

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
