# Peer review of "Evaluation of Effective Elastic Properties of Nitride NWs/Polymer Composite Materials Using Laser-Generated Surface Acoustic Waves"

_applsci, doi:10.3390/app8112319_

Reviewer 1 Report

The article is about a very important and relevant for now area of research regarding the measuring elastic properties of multilayer nanocomposites. The article is well written and I recommend it for publication in Applied Sciences.

Author Response

Dear Reviewer,

We are grateful to you for the positive estimate of our work. We have made minor corrections in the manuscript in accordance with the remarks of the second reviewer. 

Yours sincerely,

Olga Boyko

Reviewer 2 Report

Was this method of determining the effective parameters of nanocomposites tested with simpler (e.g. layer-substrate) structures, the elastic properties of which could be evaluated also by other techniques or known in advance?

How the results obtained by the authors correspond with elastic properties of single-crystal gallium nitride?

From Table 2, the drastic difference in values of some constants (e.g. C66) for samples 1 and 2 can be seen: have the authors any comments for this?

The acoustic signal shown in the insert of Fig. 4a exhibits multiple reflections. Why is this?

It should be "pulses" instead of "pulsed".in line 111, "fringes" instead of "franges" in line 112, "stronger" instead of "strong" in line 381.  

Using the term "bulk waves" is better than "body waves" (lines 288 and 292). Also, instead of a trivial formula relating the frequency to the angular frequency, the expression relating the acoustic frequency, wavelength and velocity would be more useful.   

Author Response

Dear Reviewer,

Thank you for considering our paper and for the helpful comments and remarks. Their detailed description is in the point-by-point replies below.

1. Was this method of determining the effective parameters of nanocomposites tested with simpler (e.g. layer-substrate) structures, the elastic properties of which could be evaluated also by other techniques or known in advance?

The first sample under study was silicon (substrate). The substrate is coated with nickel (very thin Ni layer (15 nm)) serving only as a transducer. The result is shown in Figure 6. From this measurement, we determined the properties of silicon used for the substrate. Figure 7 shows the measurements for extracting polymer properties, HSQ, annealed at 400 °C.

As can be seen from the article, the study of the properties of materials was carried out successively, but within the framework of one measurement technique, TGM, as noted by the reviewer.

2. How the results obtained by the authors correspond with elastic properties of single-crystal gallium nitride?

We did not restore the single-crystal properties themselves because GaN entered in the samples only as a polymer-nanowire composition (HSQ-NW). Nevertheless, we discuss the obtained results in comparison with the expected estimates based on the known properties of single GaN crystals (e.g., lines 340-351).

3. From Table 2, the drastic difference in values of some constants (e.g. C66) for samples 1 and 2 can be seen: have the authors any comments for this?

This is the result of different NW morphology (straight vs pyramidal) and amount (density) of NWs in the polymer matrix. We have intentionally chosen these two very different sample cases to show a large effect of specific NW structures on the effective constants of HSQ-NW compositions. Besides, the values of effective moduli depend on the framework of a specific mathematical model used (e.g., 3-layer or 4-layer half-space). We tried to emphasize all these aspects and recall them when discussing the numerical results obtained (e.g., in Introduction, subsections 2.1 and 4.1, lines 376-384, etc.).

4. The acoustic signal shown in the insert of Fig. 4a exhibits multiple reflections. Why is this?

This is due to the multimode nature of the excited surface waves. Judging by the two peaks in the frequency spectra, in this example, there are two SAW modes excited at the input wavelength 5.85 microns. Consequently, the transient signal in the insert shows typical beats resulted from the superposition of oscillations at these two central frequencies.

5. It should be "pulses" instead of "pulsed" in line 111, "fringes" instead of "franges" in line 112, "stronger" instead of "strong" in line 381.  

Thank you.

6. Using the term "bulk waves" is better than "body waves" (lines 288 and 292).

Corrected

Also, instead of a trivial formula relating the frequency to the angular frequency, the expression relating the acoustic frequency, wavelength and velocity would be more useful.   

Such expressions are introduced in the last paragraph of subsection 3.2 (more specifically, lines 221-225 and 229).  

Yours sincerely,

Olga Boyko